# Glutamate Decarboxylase from Lactic Acid Bacteria—A Key Enzyme in GABA Synthesis

**DOI:** 10.3390/microorganisms8121923

**Published:** 2020-12-03

**Authors:** Ida Bagus Agung Yogeswara, Suppasil Maneerat, Dietmar Haltrich

**Affiliations:** 1Food Biotechnology Laboratory, Department of Food Science and Technology, University of Natural Resources and Life Sciences BOKU, Muthgasse 18, 1190 Vienna, Austria; dietmar.haltrich@boku.ac.at; 2Nutrition Department, Faculty of Health, Science and Technology, Universitas Dhyana Pura, Dalung Kuta utara 80361, Bali, Indonesia; 3Faculty of Agro-Industry, Prince of Songkla University, Hat Yai 90110, Songkhla, Thailand; suppasil.m@psu.ac.th

**Keywords:** γ-aminobutyric acid production, lactic acid bacteria, glutamate decarboxylase, fermented foods, *gad* genes

## Abstract

Glutamate decarboxylase (l-glutamate-1-carboxylase, GAD; EC 4.1.1.15) is a pyridoxal-5’-phosphate-dependent enzyme that catalyzes the irreversible α-decarboxylation of l-glutamic acid to γ-aminobutyric acid (GABA) and CO_2_. The enzyme is widely distributed in eukaryotes as well as prokaryotes, where it—together with its reaction product GABA—fulfils very different physiological functions. The occurrence of *gad* genes encoding GAD has been shown for many microorganisms, and GABA-producing lactic acid bacteria (LAB) have been a focus of research during recent years. A wide range of traditional foods produced by fermentation based on LAB offer the potential of providing new functional food products enriched with GABA that may offer certain health-benefits. Different GAD enzymes and genes from several strains of LAB have been isolated and characterized recently. GABA-producing LAB, the biochemical properties of their GAD enzymes, and possible applications are reviewed here.

## 1. Introduction

Lactic acid bacteria (LAB) are Gram-positive, acid-tolerant, non-spore forming bacteria, with a morphology of either cocci or rods that share common physiological and metabolic characteristics. Even though many genera of bacteria produce lactic acid as their primary or secondary metabolic end-product, the term ‘lactic acid bacteria’ is conventionally reserved for genera in the order Lactobacillales, which includes *Lactobacillus*, *Lactococcus*, *Leuconostoc*, *Pediococcus*, and *Streptococcus*, in addition to *Carnobacterium*, *Enterococcus*, *Oenococcus*, *Tetragenococcus*, *Vagococcus*, and *Weisella*. LAB are important for a wide range of fermented foods and are widely used as starter cultures in traditional and industrial food fermentations [1].

Lactic acid formed during the fermentation of carbohydrates as one of the main metabolic products can affect the physiological activities of LAB. Under acidic conditions, several LAB have developed different acid-resistance systems to maintain cell viability. These systems include, for example, the F_0_F_1_-ATPase system or cation/proton antiporter/symporter systems such as K^+^-ATPase, which contribute to pH homeostasis in the cytosol by the translocation of protons [2]. In addition, glutamate or arginine-dependent systems, which require the presence of glutamate and arginine, respectively, as substrates, contribute to the acid resistance of LAB. The first enzyme in the arginine-dependent system is arginine deiminase, which degrades arginine to citrulline and NH_3_. Citrulline is then further converted to ornithine and exported from the cell by an ornithine/arginine antiporter. While the arginine-dependent system is based on the production of an intracellular alkaline compound, the glutamate-dependent system consumes an intracellular proton by combining it with internalized glutamate to γ-aminobutyric acid (GABA), and then exchanging this product for another glutamate substrate. Thereby, an extracellular amino acid is converted to an extracellular compound at the expense of an intracellular proton, which results in an increase in the intracellular pH value. This conversion of glutamate to GABA is catalyzed by glutamate decarboxylase (GAD), and the reaction requires pyridoxal-5′-phosphate (PLP) as a cofactor (Figure 1). A wide range of LAB possess the ability to produce GAD, and the biochemical properties have been studied from a number LAB sources, namely *Lactobacillus* spp., *Lactococcus* spp., and *Streptococcus* spp. [2,3]. Typically, the *gad* operon is located on the chromosomes of LAB species, with its organization varying among different species and strains [4,5,6]. Thus, GAD is important for acid resistance of LAB, but also for the formation of GABA in LAB-fermented food. GABA is the most abundant inhibitory neurotransmitter in the brain [7,8]. It has various physiological functions and is of interest as an antidepressant [9], for the induction of hypotension [10,11] and because of its cholesterol-lowering effect [12]. For example, studies by Inoue et al. and Mathieu-Pouliot et al. showed that GABA-enriched dairy products could significantly decrease the systolic blood pressure in mildly hypertensive men [10,13]. Furthermore, it was shown that GABA could prevent obesity by ameliorating oxidative stress in high-fat diet fed mice [14], and that it can effectively prevent diabetic conditions by acting as an insulin secretagogue [15,16]. Due to these properties, GABA or GABA-rich products are of interest as a food supplement or functional food.

GABA is primarily produced via different biotechnological approaches using either isolated GAD in a biocatalytic approach or various microbial strains [17], rather than through chemical synthesis due to the corrosive nature of the reactant compound [18]. GABA is currently commercialized as a nutritional supplement, however, interest in GABA-enriched food, in which GABA is formed in situ via fermentation using appropriate microorganisms, has increased lately in parallel to a general interest in functional foods. As GABA is formed as a by-product of food fermentations, LAB, which play an eminent role in the fermentation of a wide range of different products, are of particular importance when talking about GABA-enriched food. Hence, it is not surprising that strains isolated from various fermented food sources had first been shown to have the ability to produce GABA, for example, *Lactobacillus namurensis* NH2 and *Pediococcus pentosaceus* NH8 from *nham* [19], *Lactobacillus paracasei* NFRI 7415 from Japanese fermented fish [20], *L. paracasei* PF6, *Lactococcus lactis* PU1 and *Lactobacillus brevis* PM17 from cheese [21], *L. brevis* CGMCC 1306 from unpasteurized milk [22], *L. brevis* GABA100 from kimchi [23,24], *L. brevis* BJ20 from fermented sea tangle [25], *Lactobacillus futsaii* CS3 from Thai fermented shrimp [26] and *L. brevis* 119-2 and *L. brevis* 119-6 from *tsuda kabu* [12]. Recently, many studies have focused on the identification of novel GABA-producing LAB and investigated the biochemical properties of GAD from different strains in more detail [12,14,15,27,28,29].

Here, we outline the presence of *gad* genes in LAB as important and efficient GABA-producing organisms together with a phylogenetic analysis, we summarize the biochemical data available for GAD from different LAB, and finally, we give an outlook on potential applications of GAD in the manufacture of bio-based chemicals.

## 2. Biodiversity of Glutamate to γ-Aminobutyric Acid (GABA)-Producing Lactic Acid Bacteria

LAB are among the most important organisms when it comes to the fermentation of various food raw materials. They efficiently and rapidly convert sugars into lactic acid as their main metabolic product (or one of their main products), and thus contribute to the preservation of these fermented foods. Many of these raw materials or foods contain glutamate in significant amounts, which can be utilized by LAB to increase their tolerance against acidic conditions. Hence, a number of GABA-producing LAB have been isolated from a wide range of fermented foods including cheese, *kimchi*, *paocai*, fermented Thai sausage *nham*, or various fermented Asian fish products [2,13,25,26,30] (Table 1).

*Lactobacillus* spp. are the most predominant species that have been described as GABA-producing organisms including, for example, *L. brevis*, *L. paracasei*, *L. bulgaricus*, *L. buchneri,*
*L. plantarum*, *L. helveticus*, or *L. futsaii* [21,30,31,32,33,42,43]. Among these, *L. brevis,* a heterofermentative LAB, is one of the best-studied organisms [43] and is known for forming high levels of GABA under appropriate conditions (Table 1). Traditionally, fermented food samples containing GABA are used to screen for and isolate GABA-producing LAB, and it is not surprising that food samples with high GABA content may result in the isolation of promising strains showing good GABA-forming properties. Furthermore, the adjustment of the pH medium to an acidic condition (pH 4.5–5.5) could improve GABA production since GABA biosynthesis is closely related to the pH. Typical fermented foods used for isolating GABA-producing LAB are *kimchi*, where in one study, 68 out of 230 LAB isolates showed the ability to convert glutamate to GABA [44]; Thai fermented fish *plaa-som* [45], or other fermented vegetable (kimchi) [46]; fermented shrimp paste [47]; cheese [16] or milk products as well as various fermented meat or fish products including sausages or traditional fermented Cambodian food, mainly based on fish, where six out of 68 LAB isolates showed a significant GABA-producing ability [1]. These screening/isolation strategies often resulted in the identification of strains capable of efficiently converting glutamate or in the discovery of novel, not-yet-identified producers of GABA, which show promise as starter cultures for various fermented foods enriched in GABA. For example, the novel GABA producer *Lactobacillus zymae*, which can grow on up to 10% NaCl and is able to utilize D-arabitol as a carbon source, was isolated from *kimchi* [46]. Recently, Sanchart et al. isolated the novel GABA-forming strain *L. futsaii* CS3 with probiotic properties from fermented shrimp (*Kung-som*) [26,47]. This isolate was able to convert 25 mg/mL of monosodium glutamate to GABA with a yield of more than 99% within 72 h. These studies (Table 1) showed that the genera *Lactobacillus* and *Lactococcus* are the predominant GABA-producing LAB, but also other genera such as *Enterococcus* were studied in this respect. A novel GABA-producing *Enterococcus avium* strain was isolated from Korean traditional fermented anchovy and shrimp (*jeotgal*) and was shown to produce 18.47 mg/mL GABA within 48 h in a medium containing glutamate as the substrate. A recent study looking at LAB isolated from traditional Japanese fermented fish products (*kaburazushi*, *narezushi*, *konkazuke*, and *ishiru*) showed that out of 53 randomly picked LAB isolates, 10 showed the ability to transform considerable amounts of glutamate into GABA, and identified Weissella hellenica as a novel GABA producer [41]. Thus, these new genera expand the list of GABA-producing bacteria, which can open up new and different applications in the food industry. This may lead to a wider application and flexibility of starter cultures in the food industry [9]. Production of GABA by different LAB together with fermentation conditions, yields, and productivities has recently been reviewed in detail [15,43,48].

## 3. Occurrence and Organization of Glutamic Acid Decarboxylase (GAD) Genes

The conversion of glutamate to γ-aminobutyric acid is catalyzed by glutamate decarboxylase (glutamic acid decarboxylase, GAD, systematic name l-glutamate 1-carboxy-lyase (4-aminobutanoate-forming), EC 4.1.1.15), which catalyzes the irreversible α-decarboxylation of glutamate [5,48]. GAD employs pyridoxal-5′-phosphate as its cofactor, and is found in numerous microorganisms such as bacteria [3], fungi [49], and yeasts [50]; furthermore, GAD is found in plants [51], insects, and vertebrates [52]. GAD is an intracellular enzyme that is utilized by LAB to encounter acidic stress by decreasing the proton concentration in the cytoplasm in the presence of l-glutamate (Figure 2) [2,6,53,54]. This system, the so-called glutamate-dependent acid-resistance system (GDAR), provides protection under the acidic condition, and therefore the ability of LAB to perceive and cope with acid stress is crucial for successful colonization of the gastrointestinal tract (GIT) and survival under acidic environments such as in fermented food. The GDAR system consists of two homologous inducible glutamate decarboxylases, GadA and GadB, and the glutamate/γ-aminobutyrate antiporter GadC [20,48]. The corresponding genes (i.e., *gadA*, *gadB*, and *gadC*) are expressed upon entry into the stationary phase when cells are growing in rich media independently of pH, and are further induced upon hypoosmotic and hyperosmotic stress, or in the log-phase of growth in minimal medium containing glucose at a pH of 5.5 [53,55]. Siragusa et al. demonstrated that three strains with a GDAR system, *L. bulgaricus* PR1, *L. lactis* PU1, and *L. plantarum* C48, were able to survive and synthesize GABA under simulated gastrointestinal conditions [21]. Recently, cell numbers of the GABA-producing strain *L. futsaii* CS3 were shown to be only decreased by 1.5 log cycles under simulated gastrointestinal conditions, indicating that the GDAR system contributes to resistance to the conditions in the GIT and that GABA-producing LAB thus have the potential as functional probiotic starter cultures [47].

GAD systems and the organization of the *gad* operons among LAB species are highly variable [56,57]. Numerous studies reported that some LAB species such as *Streptococcus thermophilus* [5], *L. brevis* [6,7], or *L. lactis* [3] have one or two *gad* genes (i.e., *gadA*, *gadB*), together with the antiporter (*gad*C). Interestingly, *E. avium* 352 carries three *gad* genes [58]. Typically, *L. brevis* contains two GAD-encoding genes, *gadA* and *gadB*, which when expressed yield GAD enzymes that share approximately 50% amino acid sequence similarity [6]. In contrast, the *gad*B gene is absent in strain *L. brevis* CD0817 [59] and the amino acid sequence identities of GadA and GadC from *L. brevis* CD0817 against other *L. brevis* strains are 91% and 90%, respectively. The transcriptional regulator gene *gadR* plays a crucial role in GABA production and acid resistance in *L. brevis*. Gong et al. reported that deletion of *gadR* in *L. brevis* ATCC 367 resulted in lower expression of both the *gadB* and *gadC* gene, a concurrent reduction in GABA synthesis, and an increased sensitivity to acidic conditions [6]. Expression levels of *gadR* are varied among different LAB strains. The *gadR* gene was expressed 13–155-fold higher than *gadCB* in *L. brevis* NCL912 during the cultivation period [60]. In contrast, expression of *gadR* in *L. brevis* CGMCC1306 was observed to be much lower compared to *gad*CB. The role of GadA and GadB in *L. brevis* CGMCC1306 was investigated by disruption of the genes *gadA*, *gadB*, and *gadC*, resulting in complete elimination of GABA formation and increased sensitivity to acidic conditions, suggesting that both GAD proteins and the antiporter are essential for GABA production and acid resistance [61].

A genomic survey was conducted by Wu et al. to gain insight on the distribution of the *gad* operon and genes encoding glutamate decarboxylase in LAB [7]. Most strains of *L. brevis* (14 strains) as well as some strains of *L. reuteri* (six strains)*, L. buchneri* (two strains)*, L. oris* (three strains), *L. lactis* (29 strains), and *L. garvieae* (five strains) were shown to have an intact *gad* operon. The majority of these strains were shown to contain either *gadA* or *gadB*, whereas *gadC* is only present in the genomes of certain strains and noticeably lacking in *L. plantarum*, suggesting that the characteristic of GABA production is strain-dependent. Similar results were obtained by Yunes et al., who showed that *L. fermentum* (9 strains), *L. plantarum* (30 strains), and *L. brevis* (3 strains) typically contain *gad*B genes. In addition, no antiporter gene was observed next to *gad*B in *L. plantarum* 90sk, and the expression of *gad*B was increased in the early stationary phase and at low pH (3.5–5) [62]. The *gad*B gene from *S. thermophilus* encoding 459 amino acids has been investigated. The transposase genes Tn1216 (5′ and 3′) and Tn1546 are located downstream and upstream of hydrolase genes flanking the *gad*B/*gad*C operon as a result from horizontal gene transfer. This sequence implies that the order of *gad*B and *gad*C in *S. thermophilus* ST110 is similar to *S. thermophilus* Y2 [63], but in a different order from that reported for *L. lactis* [64], *L. brevis* [60], and *L. plantarum* [62].

The *L. reuteri* 100-23 genome was investigated by Su et al. for its *gad* operon [65]. This genome contains *gad*B and two genes for the antiporter (*gadC1* and *gadC2*) as well as the glutaminase-encoding gene *gls3*, indicating that glutamine serves as a substrate for the synthesis of GABA. The organization of the *gad* operon is in a different order for other species of LAB (*L. lactis* and *L. plantarum*) as glutaminase (*gls3*) is in between the antiporters *gadC1* and *gadC2*, while *gad*B is accompanied by *gad*C1 [65]. The full length of *gad* genes has been cloned and sequenced for several species and strains of LAB. Li et al. cloned *gadA* from *L. brevis* NCL912, and the whole gene fragment (4615 bp) including *gadR*, *gadC*, *gadA*, and *gts* (glutamyl t-RNA synthetase) was successfully amplified. Their work suggested that the high GABA production capacity of *L. brevis* NCL912 may be linked to the *gadA* locus, forming a *gadCA* operon complex that ensures the coordinated expression of GAD and the antiporter [60]. A core fragment of the *gad* gene from *L. brevis* OPK3 was cloned and successfully expressed in *Escherichia coli*. The nucleotide sequence revealed that the open reading frame of the *gad* gene consisted of 1401 bases encoding 467 amino acid residues. The sequence showed 83%, 71%, and 60% homology to GAD from *L. plantarum*, *L. lactis*, and *Listeria monocytogenes,* respectively [66].

A phylogenetic tree constructed from available GAD sequences in the NCBI protein database showed that amino acid sequences of GAD are highly conserved within the same species (Figure 3), and that GAD is widely distributed in a number of LAB including *L. brevis, L. buchneri, L. delbrueckii* subsp. *bulgaricus, L. fermentum, L. futsaii, L. paracasei, L. parakefiri, L. paraplantarum, L. plantarum, L. plantarum* subsp. *argentoratensis, L. reuteri, L. sakei, L. lactis*, and *S. thermophilus*. All of these LAB are commonly found in fermented foods and some of these are commonly used as starter cultures in food industries. In addition, GAD is also found in other lactobacilli including *L. acidifarinae, L. aviaries, L. coleohominis, L. farraginis, L. japonicas, L. koreensis, L. nuruki, L. oris, L. rossiae, L. rennini*, or *L. suebicus* (Figure 3). These organisms have not been studied for their capacity to synthesize GABA nor have their GAD system been studied, and hence they could be of interest with respect to GABA production and GABA-enriched food.

## 4. Glutamate Decarboxylase

Glutamate decarboxylase is an intracellular enzyme that is found ubiquitously in eukaryotes and prokaryotes. GAD exhibits different physiological roles, especially in vertebrates and plants, and its presence is highly variable among organisms [52]. GAD is a PLP-dependent enzyme and as such belongs to the PLP-dependent enzyme superfamily. This superfamily comprises seven different folds [67] with GAD from LAB showing the type-I fold of PLP-dependent enzymes [68]. A number of important catalytic reactions including α- and β-eliminations, decarboxylation, transamination, racemization, and aldol cleavage are catalyzed by various members of this superfamily of enzymes [69]. GAD activity relies on the binding of its co-factor PLP, and belongs to group II of PLP-dependent decarboxylases [70]. In GAD from *L. brevis* CGMCC 1306, the active site entrance is located at the *re*-face of the cofactor PLP. PLP is covalently attached to a lysine (K279) via an imine linkage (Figure 4), referred to as an internal aldimine [68,71]. This lysine is strictly conserved in group II PLP-dependent decarboxylases. The corresponding lysine in *E. coli* GAD is at position 276, and when mutating this residue, the variant has less flexibility and affinity to both its substrate and the cofactor [72]. In addition to this covalent attachment, PLP is positioned in the active site via a number of H bonds between the phosphate group of PLP and surrounding amino acids, while the pyridine ring of PLP forms hydrophobic interactions with side chains of various amino acids in the active site [68].

Molecular docking of the substrate glutamate into the active-site of the holo-form of *L. brevis* GAD showed several noncovalent interactions including hydrogen bonds between the O2, the O3 and the O4 atoms of the substrate L-Glu to various parts of the GAD polypeptide chain. Furthermore, electrostatic interactions between the negatively charged oxygen atom of the α-carboxyl and the γ-carboxyl group of L-Glu and the positively charged nitrogen atom of residue R422 as well as H278 and K279 (Figure 5), respectively, were proposed [68]. The flexible loop residue Tyr308-Glu312 in *L. brevis* GAD is located near the substrate-binding site (Figure 4). This loop is important for the catalytic reaction, and the conserved residue Tyr308 plays a crucial role in decarboxylation of L-Glu. Thr 215 and Asp246 are the two catalytic residues in *L. brevis* GAD (Figure 5), which are also highly conserved and promote decarboxylation of L-Glu [68,71,73].

During catalysis, a transamination reaction occurs, and PLP, which is covalently attached to a Lys in the active site of GAD in its resting state, now becomes covalently bonded to the substrate glutamate, forming a Schiff base or what is referred to as an external aldimine. This Schiff base can then be transformed to a quinonoid intermediate [67,74]. In a small fraction of catalytic cycles, when glutamate is decarboxylated, a subsequent alternative transamination of the quinonoid intermediate of the reaction can occur, and succinic semialdehyde (SSA) and pyridoxamine-5′-phosphate (PMP) are formed. The latter will immediately be released from the enzyme, resulting in inactive apoGAD (Figure 6), which can be regenerated to the active GAD–PLP complex when free pyridoxal-5′-phosphate is present, thus completing a cycle of inactivation and activation. However, when free PLP is not present, GAD will be inactivated as a function of time and substrate concentration [62,67,68,69,74,75,76,77].

## 5. Biochemical Insights into Glutamate Decarboxylase from Lactic Acid Bacteria

GAD from LAB typically consists of identical subunits with molecular masses ranging from 54 to 62 kDa and is formed in its mature holo-form, even when produced heterologously. The oligomerization, typically resulting in the formation of a homodimer, is crucial for activity of the *Lactobacillus* spp. enzymes. Some ambiguity about the active form of GAD isolated from different isolates of *L. brevis* and its quaternary structure exists in the scientific literature. Hiraga et al. reported that treatment with high concentrations of ammonium sulfate resulted in an active tetrameric form with the enzyme from *L. brevis* IFO12005 GAD [78]. The presence of ammonium sulfate apparently stabilized GAD from this source as the purified enzyme was found to be rather unstable, and the dimeric form showed no activity. Moreover, the presence of ammonium sulfate apparently did not affect the overall conformation but had effects on the active site of the protein. Studies by Yu et al. showed that GAD from *L. brevis* CGMCC 1306 is active as a monomer, while GAD from other LAB are generally active as dimers [71]. Subsequent structural studies on this enzyme revealed, however, that GAD from *L. brevis* CGMCC 1306 is active as a dimer (Figure 7), even though elucidation of the crystal structure resulted in a distorted asymmetric trimer. The authors concluded that this observed trimer only resulted from the crystallographic packing and not the biological form [68].

As above-mentioned, a number of LAB carry two GAD-encoding genes, *gadA* and *gadB*. Frequently, studies have focused on the purification and characterization of GadB (e.g., from *L. plantarum* [79], *L. sakei* [80], *L. brevis* [78], *Enterococcus raffinosus* [75], and *L. paracasei* [18]), since the expression levels of recombinant GadB are typically higher than those for GadA [55]. A recent study by Wu et al. showed that the *gadA* transcript was highly upregulated (55-fold) in strain *L. brevis* NPS-QW-145 at the stationary phase of growth [7]. Subsequently, both GadA and GadB were recombinantly produced and characterized. GadA showed a pH profile of activity near the neutral region, with the optimal activity found in the range of pH 5.5–6.6, in contrast to GadB, which is more active under acidic conditions (3.0–5.5). Presence of both of these two enzymes, GadA and GadB, in the *L. brevis* genome will give the organism a significant advantage to produce GABA over a broad range of pH (3.0–6.0), and thus to more efficient maintenance of pH homeostasis. These findings suggest that extending the activity of GadA to the near-neutral pH region offers a novel genetic diversity of *gad* genes from LABs [7].

A number of GAD have been expressed and characterized from a variety of LABs. In general, the N- and C-terminal regions of GAD from different sources show significant differences, and this might affect recombinant GABA production. As shown in a sequence alignment (Figure 8), the sequence HVD(A/S)A(S/F)GG was highly conserved among LAB GAD, and a lysine residue (Lys279 in *L. brevis* GAD) played a crucial role in the PLP binding site. Table 2 summarizes the biochemical properties of GAD from different strains [18,42,81,82]. Typically, the pH optima of GAD are found between 4.0 and 5.0. GAD from *L. zymae*, *E. avium* M5, *S. salivarius* subsp. *thermophilus* Y2, and *L. paracasei* NFRI 7415 have an optimum activity of above 40 °C, which does not coincide with the optimal temperature for growth of these strains [46,72,82,83]. Different ions can affect the stability and activity of GAD from different sources (Table 2). GAD from *E. avium* M5 is activated in the presence of CaCl_2_ and MnCl_2_ but the activity is decreased by CuSO_4_ and AgNO_3_ [82]; comparable results were also obtained for GAD from other LAB sources, *L. zymae* [46] and *L. sakei* A156 [80].

Since GAD is mainly active under acidic conditions, several engineering approaches have been employed to broaden its activity, especially at the near-neutral pH region. To this end, Shi et al. applied both directed evolution and site-directed mutagenesis at the β-hairpin region and *C*-terminal end of *L. brevis* GAD [84]. By using a plate-based screening assay employing a pH indicator as assay principle, they could identify several variants and positions that improved activity at pH 6.0. Furthermore, they selected three residues (Tyr308, Glu312, Thr315) in the β-hairpin region for site-directed mutagenesis based on homology modeling, since these residues exhibit different interactions with surrounding amino acids in the model at different pH values. By combining various positive mutations, they could increase the catalytic efficiency of GAD from *L. brevis* 13.1- and 43.2-fold at pH 4.6 and 6.0, respectively, when compared to the wild-type enzyme [84]. The role of the *C*-terminus for the pH dependence of catalysis of *L. plantarum* GAD was investigated by Shin et al. employing mutagenesis [79]. Deletions of three and eleven residues in the C-terminal region Ile454-Thr468 of this enzyme increased activity in the pH range of 5 to 7, with the Δ11 variant showing significantly better results, increasing the catalytic efficiency of the variant at pH 5.0 and 7.0 by a factor of 1.26 and 28.5, respectively. The authors concluded that the *C*-terminal region is involved in decreasing the activity of *L. plantarum* GAD at higher pH values by closing up the catalytic site as a result of pH-induced conformational changes [79]. In a similar way, a *C*-terminally truncated variant of *L. brevis* GAD, in which the terminal 14 amino acids had been removed by site-directed mutagenesis, showed improved activity at higher, around neutral pH values [85]. These studies point to the importance of the *C*-terminus of GAD for improved accessibility of the active site and increased activity, especially at higher pH values, and thus the *C*-terminal loop is an essential target for enzyme engineering for GABA production at fluctuated pH conditions [79,85].

## 6. Improvement of GAD Activities and GABA Production

GABA biosynthesis can be achieved by using whole cell reactions, recombinant bacteria, and purified GAD (Table 3). *gad* genes from various sources of LAB have been overexpressed in different hosts including *E. coli* [86], *L. sakei* [87], *L. plantarum* [88], *Corynebacterium glutamicum* [89], and *Bacillus subtilis* [90]. Utilization of whole cells for the biocatalytic conversion of glutamate to GABA has some drawbacks including the conversion of GABA to succinic semialdehyde by the enzyme GABA transaminase (GABA-T), which is often found in bacteria and might decrease GABA yields during cultivation. To prolong and thereby increase GABA production, continuous cultivation [91], fed-batch fermentation [92] as well as immobilized cell technology [93,94,95] have been employed. All of these approaches effectively increased GABA productivity by improving cell viability resulting in extended periods of cultivation.

GABA biosynthesis and production could be enhanced by optimizing fermentation conditions, with attention given to different factors including the carbon source, concentration of added glutamate, pH regulation, incubation temperature, nitrogen sources, cofactor, and feeding time [34,94]. A study by Lim et al. showed that under optimized conditions, *L. brevis* HYE1 produced 18.8 mM of GABA. Monosodium glutamate (MSG) or l-glutamate are the main substrate for the production of GABA using either appropriate GAD-containing cells or pure GAD [27]. LAB with GAD activity may furthermore require the supplementation of PLP to the medium to enhance GABA production. The addition of 0.5% MSG increased GABA production by *E. faecium* JK29, which reached 14.9 mM after 48 h of cultivation [38]. A concentration of 6% MSG and the addition of 0.02 mM PLP were found to be optimal conditions for *L. brevis* K203 for GABA production [42]. This strategy of increasing glutamate supplementation could not be used for all strains though; when l-glutamate was added at concentrations of 10 to 20 g/L to the growth medium of *S. thermophilus*, GABA production could not be enhanced. It was suggested that this strain is not able to tolerate high glutamate concentrations [36]. High glutamate concentrations increase the osmotic pressure in the cells, and this stress can disturb the bacterial metabolism [39]. Fermentation time and temperature are also key factors for GABA production. Villegas et al. investigated GABA formation by *L. brevis* CRL 1942, and found that 48 h of fermentation at 30 °C employing 270 mM of MSG resulted in a maximum GABA production of 255 mM in MRS medium, indicating that the GABA production occurs in a time-dependent manner [96].

Metabolic pathway engineering has been performed to achieve enhanced GABA production. The key points here are the direct modulation of GABA metabolic pathways. A whole-cell biocatalyst based on *E. coli* cells expressing the *gad*B gene from *L. lactis* was used as the starting point of this engineering approach. An engineered strain was constructed by (i) introducing mutations into this GadB to shift its decarboxylation activity toward a neutral pH; (ii) by modifying the glutamate/GABA antiporter GadC to facilitate transport at neutral pH; (iii) by enhancing the expression of soluble GadB through overexpression of the GroESL molecular chaperones; and (iv) by inhibiting the degradation of GABA through inactivation of *gadA* and *gadB* from the *E. coli* genome. This engineered strain achieved a productivity of 44.04 g/L of GABA per h with an almost quantitative conversion of 3 M glutamate [97].

Several mutational approaches such as directed evolution and site-specific mutagenesis are considered as powerful tools for optimizing or improving enzyme properties. Several researchers have applied these approaches to improve GAD activity [84,97,98,99,100,101] and were applied in whole-cell biocatalysts. In order to improve GAD activity over an expanded pH range, recombinant *C. glutamicum* cells were obtained by expressing *L. brevis* Lb85 GadB variants. These variants were constructed by combining directed evolution and site-specific mutagenesis of GadB to improve activity at higher pH values (see above), since *C. glutamicum* grows best around neutral pH [84]. *C. glutamicum* is an industrial producer of glutamate, and by introducing these GadB variants into this organism, GABA could be produced without the need of exogenous glutamate on a simple glucose-based medium, with yields of up to 7.13 g/L [84].

Insufficient thermostability is often a common problem associated with industrial enzymes, and most GAD show low stability even at moderate temperatures. A rational strategy for improving thermostability is to identify critical regions or amino acid residues by sequence alignments. Alternatively, structural information indicating flexible regions can be used, and subsequently, these regions are strengthened [102]. Identification of the consensus sequences can also improve the thermostability of proteins [103]. Recently, Zhang et al. developed a parallel strategy to engineer *L. brevis* CGMCC 1306 GAD. They compared the sequence and structure of this mesophilic GAD with homologous thermophilic enzymes to identify amino acid residues that might affect stability. Two mutant enzymes were obtained and showed higher thermostability with their half-inactivation temperature 2.3 °C and 1.4 °C higher than that of the wild-type enzyme. Furthermore, the activity of the variants was 1.67-fold increased during incubation at 60 °C for 20 min. They suggested that this approach can be an efficient tool to improve the thermostability of GAD [102].

The use of purified GAD seems to be economically more feasible than whole-cell biocatalysis when aiming at producing pure GABA due to simplified downstream purification of this compound from less complex reaction mixtures. A number of immobilization techniques have been applied for re-use of the biocatalyst such as immobilization of GadB in calcium alginate beads that are then employed in a bioreactor [104], a GAD/cellulose-binding domain fusion protein immobilized onto cellulose [105], and GAD immobilized to metal affinity gels [106]. The performance of immobilized GAD in a fed-batch reactor was evaluated, which showed high productivity of GABA as the substrate concentration in the medium was kept constant by feeding solid glutamate. Moreover, no significant decrease in enzyme activities was observed during the reaction when the inactivation reaction of PLP to succinic semialdehyde and pyridoxamine-5′-phosphate during catalysis was avoided by the addition of a small amount of α-ketoglutaric acid to the reactor, which regenerated PLP [101]. Sang-Jae Lee et al. performed immobilization of *L. plantarum* GAD using silica beads and showed high stability under acidic and alkaline conditions with improved thermostability [105]. In addition, the immobilized GAD converted 100% of glutamate to GABA [106]. These results suggest that immobilization gives advantageous results for industrial application when using (partially) purified GAD for GABA production from glutamate.

## 7. The Role of Glutamate Decarboxylase in the Manufacturing of Bio-Based Industrial Chemicals

Agricultural waste and waste streams from biofuel production are now being considered as a low-cost source of glutamate for biotechnological conversion into GABA and production of bio-based chemicals [107]. These protein-rich materials are mainly bioethanol by-product streams including dried distiller’s grains with solubles (DDGS) from maize and wheat, or vinasse from sugarcane or sugar beet, but also plant leaves, oil, or biodiesel by-products and slaughterhouse waste. In the future, algae could also provide an additional source for biodiesel and thus become a natural low-cost source of glutamic acid.

The protein-rich fraction of plants can be further split into more- and less-nutritious fractions, for example, by hydrolyzing the proteins and separating the essential (nutritious) amino acids from the non-essential (less nutritious) ones. Non-essential amino acids such as glutamic acid and aspartic acid, which have no significant value in animal feed, can be utilized for preparing functionalized chemicals. Recently, a by-product from the tuna canning industry, tuna condensate, was shown to be a useful material for the production of GABA. Tuna condensate contains significant amounts of glutamine, but relatively little glutamate. Glutamine was first converted to glutamate by a glutaminase from *Candida rugosa*, and in a second step, *L. futsaii* GAD converted glutamate to GABA. Both steps were catalyzed by immobilized whole cells [108]. Recently, it was shown that supplementation of arginine to media containing glutamate could enhance GABA production, and that the simultaneous addition of arginine, malate, and glutamate enabled GABA production already during exponential growth at relatively high pH (6.5) [109].

The structure of glutamic acid resembles many industrial intermediates, so it can be transformed into a variety of chemicals using a relatively limited number of steps. Decarboxylation of glutamic acid to GABA, enzymatically performed by GAD, is an important reaction of the pathway from glutamic acid to a range of molecules. GABA is, for example, an intermediate for the synthesis of pyrrolidones. Such an approach can be used to produce *N*-methyl-2-pyrrolidone (NMP), which is used as an industrial solvent. Combining the enzymatic decarboxylation of glutamate performed by GAD with the one-pot cyclization of GABA to 2-pyrrolidone and subsequent methylation will thus yield NMP [110]. Another interesting material synthesized by ring-opening polymerization of 2-pyrrolidone is Nylon 4 [111], a four-carbon polyamide suitable for application as an engineering plastic due to its superior thermal and mechanical properties [112]. Contrary to other nylon polymers, Nylon 4 is heat-resistant, biodegradable, biocompatible, and compostable [112].

## 8. Future Trends and Conclusions

The demand for functional foods is increasing and marked by the awareness of consumers in maintaining health and prevention of degenerative diseases. Therefore, exploration of bioactive compounds such as GABA are important. The GAD system plays a crucial role in GABA biosynthesis. A number of studies on cloning, expression, and characterization of both *gadA* and *gadB* and the encoded enzymes GadA and GadB has led to deciphering the role of the *gad* genes in the GABA metabolic pathway and its importance for LAB. Since the production of GABA is dependent on the biochemical properties of GAD, more study on the biochemical properties of GAD are important, especially for those enzymes derived from LAB isolated from food fermentation processes, as this will facilitate the optimization of the fermentation process and support the selection of suitable starter cultures for these processes that will bring more GABA-enriched food to the consumer. Recent structural information of GAD from LAB will facilitate enzyme-engineering approaches to improve GAD toward enhanced thermostability or improved activity over a broad range of pH. However, structural information is currently only limited to GAD from *L. brevis*, and thus structural studies on GAD from other GABA-producing LAB are needed in order to understand their catalytic and structural properties in more depth. The elucidation of molecular mechanisms and roles of GABA production, knowledge of the regulatory aspects of GABA production, and profound comprehension of GABA-producing cell physiology will offer the basis and tools to increase GABA yields at genetic and metabolic levels.

## Figures and Tables

**Figure 1 microorganisms-08-01923-f001:**
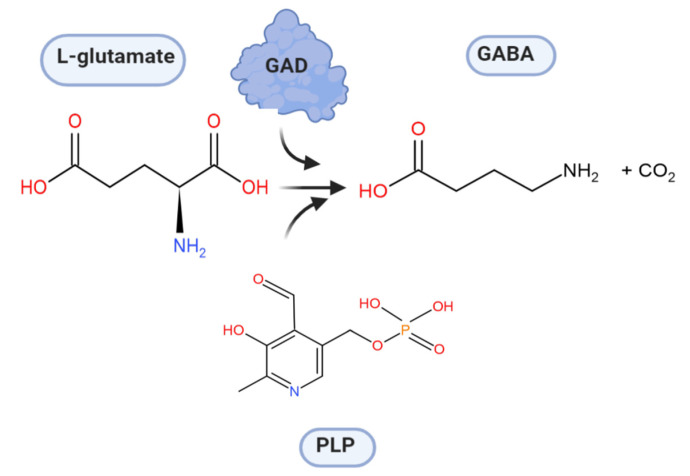
Decarboxylation of l-glutamate to GABA catalyzed by glutamate decarboxylase. PLP: pyridoxal-5′-phosphate.

**Figure 2 microorganisms-08-01923-f002:**
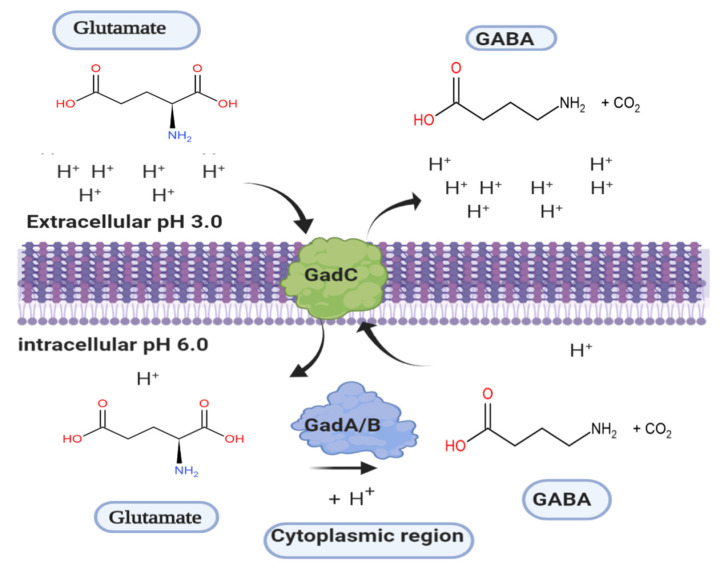
Schematic representation of the glutamate-dependent acid-resistance system. Glutamate (net charge 0) is taken up by the l-glutamate/GABA antiporter GadC, while concurrently GABA is exported by GadC as indicated by the arrows. Subsequently, GadA/B catalyze the decarboxylation of glutamate by consuming an intracellular proton (H^+^) at each cycle and generate the proton motive force by GABA export (net charge +1).

**Figure 3 microorganisms-08-01923-f003:**
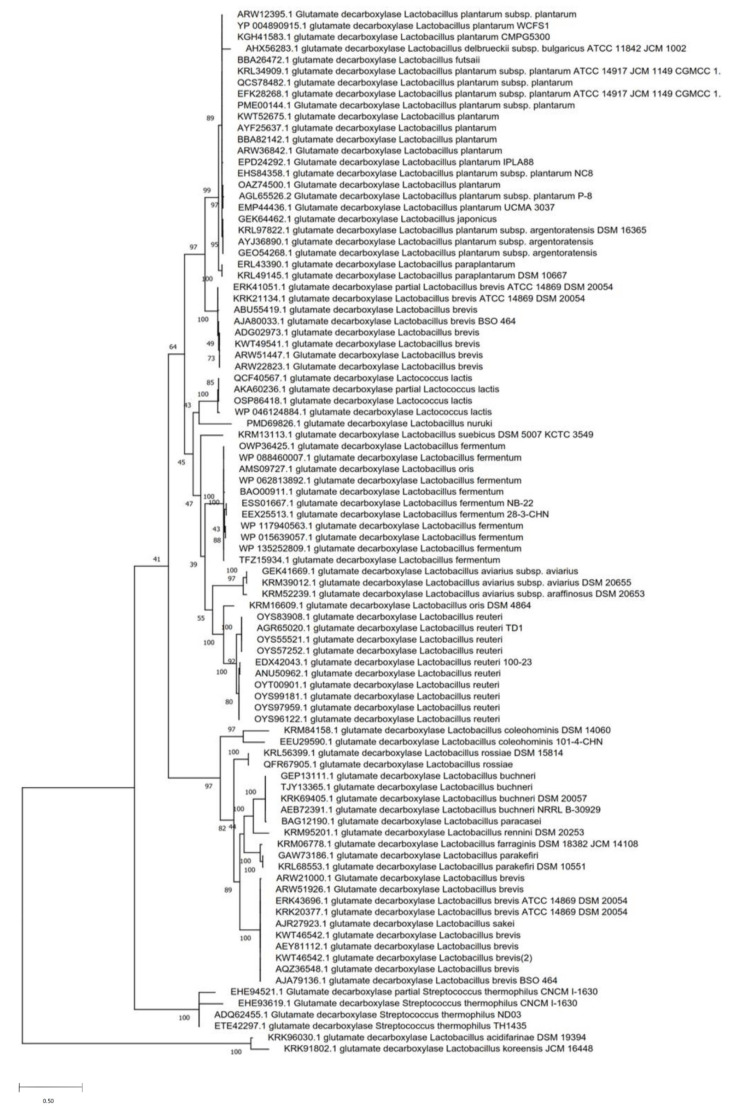
Phylogenetic analysis of glutamate decarboxylase from different species of LAB. The phylogenetic tree was calculated based on the amino acid sequences of glutamic acid decarboxylase (GAD) (maximum-likelihood method). The phylogenetic analysis was performed after the alignment of GAD sequences using MUSCLE in the MEGA X software.

**Figure 4 microorganisms-08-01923-f004:**
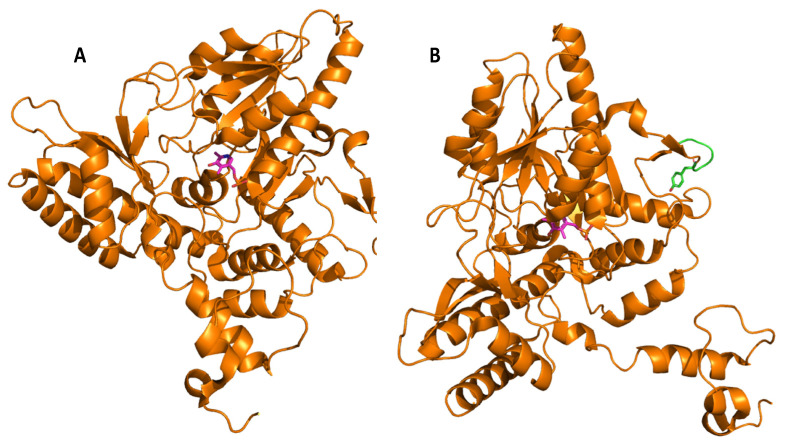
Overall secondary structure of the glutamate decarboxylase monomer from *L. brevis* (PDB code 5GP4). (**A**) Chain A is represented as an orange cartoon, and its prosthetic group PLP is represented as sticks colored by atom type, with carbons shown in magenta. (**B**) Position of the Y308-E312 flexible loop shown in green. The conserved Y308 is represented as sticks, colored by atom types, with carbons being green. All images were made using the PyMOL Molecular Graphics System, v. 2.3.0. for Linux.

**Figure 5 microorganisms-08-01923-f005:**
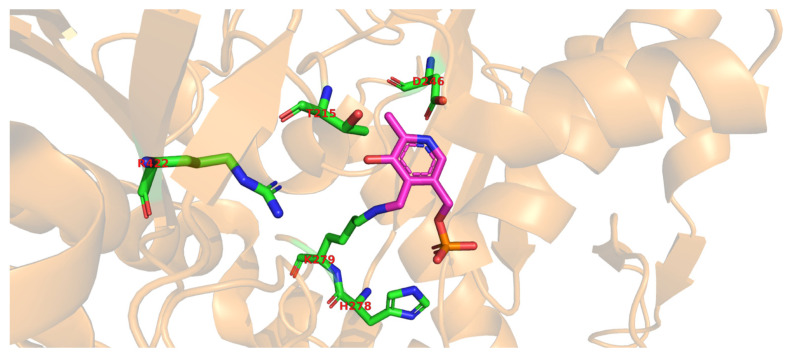
Active site of glutamate decarboxylase from *L. brevis* (PDB code 5GP4). The conserved catalytic residues T215 and D246 are shown in sticks, colored by atom type, with carbons shown in green, nitrogen in blue, and oxygen in red. The prosthetic group PLP is represented as sticks colored by atom type with carbons in magenta. The residues R422, H278, and K279, proposed to be involved in electrostatic interactions with the substrate glutamate [68], are represented as sticks colored by atom type. The rest of the chain is shown as a transparent orange cartoon. K279 is also involved in forming the imine linkage to PLP. The image was made using the PyMOL Molecular Graphics System, v. 2.3.0. for Linux.

**Figure 6 microorganisms-08-01923-f006:**
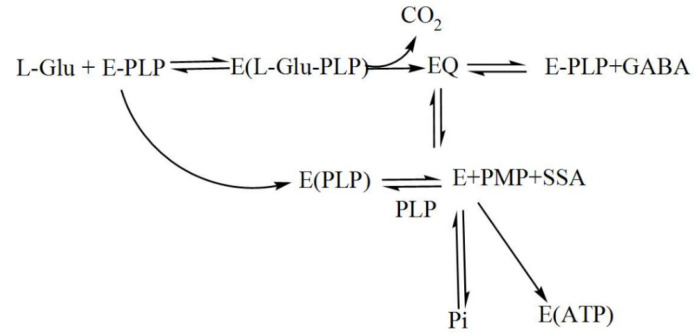
The interconversion of holo- and apoGAD. The primary reaction results in the formation of GABA and holoGAD remains intact and active. holoGAD reacting with PLP will activate a secondary reaction resulting in the formation of apoGAD. E, apoGAD; E-PLP, holoGAD; Pi, inorganic phosphate; EQ, quinonoid intermediate; PMP, pyridoxamine phosphate; PLP, pyridoxal-5′-phosphate; SSA; succinic semialdehyde (modified from [76]).

**Figure 7 microorganisms-08-01923-f007:**
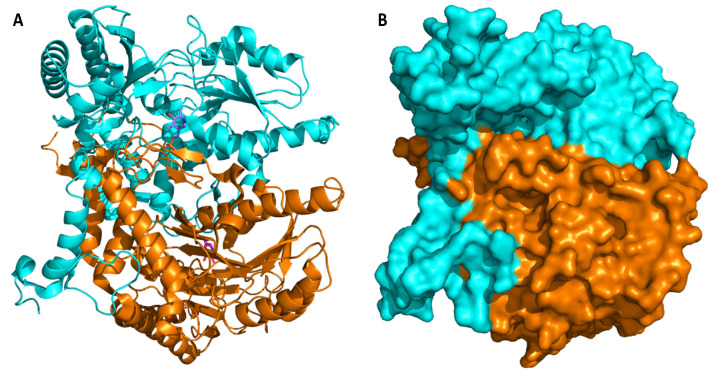
Structure of homodimeric glutamate decarboxylase from *L. brevis* CGMCC 1306 (PDB code: 5GP4). (**A**): overall secondary structure with chains A and B represented as orange and cyan cartoons, respectively. The prosthetic group PLP is represented as sticks, colored by atom type, with carbons shown in magenta or blue. (**B**). Surface of GAD with chains A and B of the crystal structure represented as orange and cyan surfaces, respectively. Images were made using the PyMOL Molecular Graphics System, v. 2.3.0. for Linux.

**Figure 8 microorganisms-08-01923-f008:**
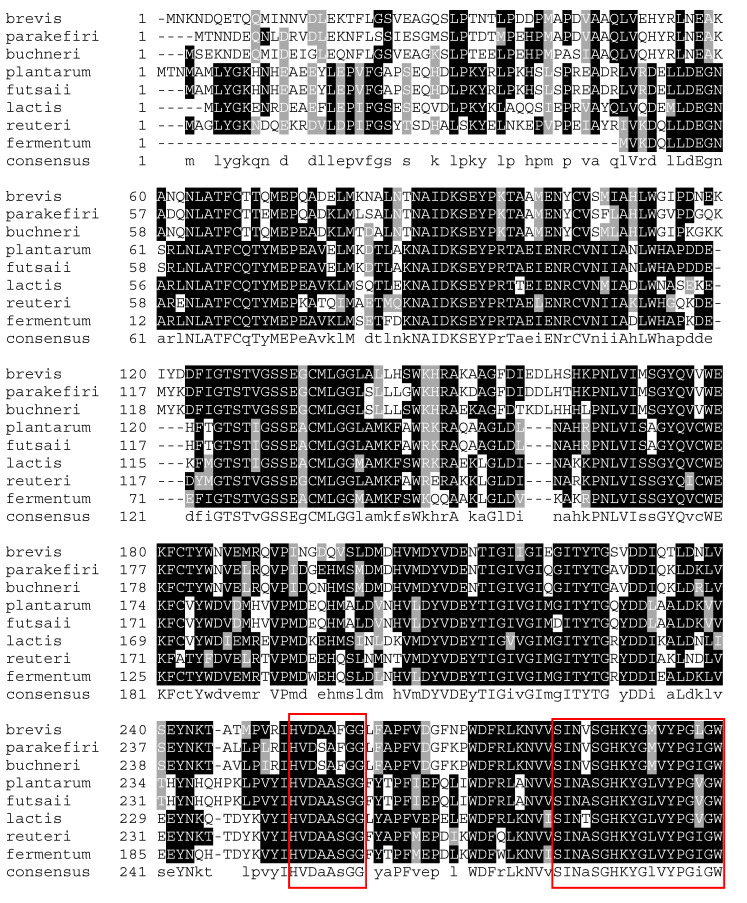
Comparison of amino acid sequences of GAD from *L. brevis*, *L. parakefiri*, *L. buchneri*, *L. plantarum*, *L. futsaii*, *L. lactis*, *L. reuteri*, and *L. fermentum*. The accession numbers of these sequences are GAW73186.1, ERK43696.1, KRL34909.1, AEB72391.1, OYT00901.1, ESS01667.1, BBA26472.1, and OSP86418.1, respectively. The alignment of amino acids was generated using the Clustal Omega software. The boxed sequence indicates residues HVD(A/S)A(S/F)GG; this sequence is highly conserved in PLP-dependent decarboxylases [48,55]. Furthermore, the residues SINA/V/TSGHKYGM/LVYPGI/V/LGWI/VV/LW/R/K/V are part of the PLP-binding domain [26].

**Table 1 microorganisms-08-01923-t001:** Diversity of glutamate to γ-aminobutyric acid (GABA)-converting lactic acid bacteria (LAB), isolation sources, GABA production, and fermentation conditions. GABA concentrations as found in food products fermented with this strain are given.

LAB Species and Strain	Sources	Fermentation Conditions	GABA Production	References
*L. brevis* HY1	*Kimchi*	30 °C, 48 h	18.76 mM	[27]
*L. brevis* NCL912	*Paocai*	pH 5.0, 32 °C, 36 h Fed-batch fermentation	149.05 mM	[28]
*L. helveticus* NDO1	*Koumiss*	pH 3.5, 30 °C, 30 h	0.16 g/L	[29]
*L. brevis* BJ20	Fermented *jotgal*	30 °C, 24 h	2.465 mg/L	[25]
*L. paracasei* 15C	Raw milk cheese	pH 5.5, 30 °C, 48 h, anaerobe	14.8 mg/kg	[30]
*L. rhamnosus* 21D-B	Raw milk cheese	pH 5.5, 30 °C, 48 h, anaerobe	11.3 mg/kg	[30]
*S. thermophilus* 84C	Raw milk cheese	pH 5.5, 30 °C, 48 h, anaerobe	80 mg/kg	[30]
*L. plantarum* DM5	*Marcha Sikkim*	pH 6.4, 30 °C, 30 h	NR	[31]
*L. brevis* L-32	*Kimchi*	30 °C, 24 h	38 g/L	[32]
*L. buchneri* WPZ001	Chinese fermented sausage	30 °C, 72 h	129 g/L	[33]
*L. lactis*	*Kimchi*	pH 5.5, 30 °C, 20 h	6.41 g/L	[34]
*L. otakiensis*	Pico cheese	30 °C, 48 h	659 mg/L	[35]
*S. thermophilus* Y2	Yoghurt	pH 4.5, 40 °C, 100 h	7.98 g/L	[36]
*L. buchneri* MS	*Kimchi*	pH 5.0, 30 °C, 36 h	251 mM	[37]
*E. faecium* JK29	*Kimchi*	30 °C, 72 h	14.86 mM	[38]
*L. brevis* 877G	*Kimchi*	30 °C, 24 h	18.94 mM	[39]
*L. plantarum* IFK 10	fermented soybean	pH 6.5, 37 °C, 48 h	2.68 g/L	[40]
*Weissella hellenica*	*ika-kurozukuri*	30 °C, 96 h	7.69 g/L	[41]
*L. brevis* K203	*Kimchi*	pH 5.25, 37 °C, 48 h	44.4 g/L	[42]
*L. futsaii* CS3	*Kung-som*	37 °C, 108 h	25 g/L	[26]
*L. paracasei* NFR7415	Fermented fish	30 °C, 144 h	302 mM	[20]
*L. plantarum* C48	Cheese	30 °C, 48 h	16 mg/kg	[21]
*L. paracasei* PF6	Cheese	30 °C, 48 h	99.9 mg/kg	[21]
*L. brevis* PM17	Cheese	30 °C, 48 h	15 mg/kg	[21]
*L. lactis* PU1	Cheese	30 °C, 72 h	36 mg/kg	[21]
*L. delbrueckii* subsp. *bulgaricus* PR1	Cheese	42 °C, 48 h	63 mg/kg	[21]
*L. lactis* subsp. *lactis*	Cheese starter	30 °C, 48 h	27.1 mg/L	[3]
*L. brevis* CECT 8183	Goat cheese	pH 4.7, 30 °C, 48 h	0.96 mM	[16]
*L. brevis* CECT 8182	Sheep cheese	pH 4.7, 30 °C, 48 h	0.94 mM	[16]
*L. brevis* CECT 8182	Goat cheese	pH 4.7, 30 °C, 48 h	0.99 mM	[16]
*L. lactis* CECT 8184	Goat cheese	pH 4.7, 30 °C, 48 h	0.93 mM	[16]
*L. namurensis* NH2	*Nham*	30 °C, 24 h	9.06 g/L	[17]
*P. pentosaceus* HN8	*Nham*	30 °C, 24 h	7.34 g/L	[17]
*L. plantarum*	*paork kampeus*	pH 6.5, 37 °C, 72 h	20 mM	[1]

NR: Not reported.

**Table 2 microorganisms-08-01923-t002:** Biochemical properties of glutamate decarboxylase from various LAB.

Source	Molecular Mass of Subunit (kDa)	Optimal pH	Optimal Temperature	Effect of Metal Ions (Increased Activity)	Effect of Metal Ions (Decreased Activity)	K_m_ (Mm)	V_max_	References
*L. zymae*	53	4.5	41	NH_4_^+^, Ca^2+^, Mg^2+^, Mn^2+^, Na^+^	Co^2+^, Cu^2+^, Ag^+^	1.7	0.01 mM/min	[46]
*L. paracasei* NFRI 7415	57	5	50	NH_4_^+^, Ca^2+^	EDTA, Na^+^	5	NR	[18]
*L. sakei* A156	54.4	5	55	Mn^2+^, Co^2+^, Ca^2+^, Zn^2+^	NH_4_^+^, Mg^2+^, Ag^+^	0.045	0.011 mM/min	[80]
*L. brevis* CGMCC 1306	53	4.8	48	NR	NR	10.26	8.86 U/mg	[22]
*S. salivarius* subsp. *thermophilus Y2*	46.9	4	55	Ba^2+^	Fe^2+^, Zn^2+^, Cu^2+^, Mn ^2+^, Na^+^, Ag^+^, Co^2+^, Li^+^, K^+^	2.3	NR	[83]
*Enterococcus avium* M5	53	4.5	55	Ca^2+^, Mg^2+^, Mn^2+^, Zn^2+^	Cu^2+^, Ag^+^	3.26	0.012 mM/min	[82]
*E. raffinosus* TCCC11660	55	4.6	45	Mo^6+^, Mg^2+^	Fe^2+^, Zn^2+^, Cu^2+^, Co^2+^	5.26	3.45 µM/min	[75]
*Lactococcus lactis*	NR	4.7	NR	NR	NR	0.51	NR	[3]
*L. brevis* 877G	50	5.2	45	Ca^2+^, Mg^2+^, Mn^2+^, Na^+^	Ag^+^, Zn^2+^, Cu^2+^, K^+^	3.6	0.06 mM/min	[81]

NR; Not reported.

**Table 3 microorganisms-08-01923-t003:** Various approaches to improve GABA production.

Strain	GABA Enhancement Techniques	Reaction Conditions	GABA Production	References
*L. plantarum* Taj-apis 362	GAD was expressed in pMG36e vector	Resting cells, reaction mixtures contain 1.32 mM glutamic acid and 200 mM sodium acetate, incubated at 37 °C for 60 min	1.14 g/L	[88]
*L. plantarum* ATCC 14917	*L. sakei* expression host	MRS supplemented with 1% MSG, incubated at 30 °C for 48 h, initial pH 6.0	27.36 g/L	[87]
*S. salivarius* ssp, *thermophilus* Y2	*B. subtilis* expression host	Resting cells, reaction mixtures contain 0.4 M sodium glutamate and 0.4 M acetate buffer, incubated at 37 °C for 6 h	5.26 g/L	[90]
*L. brevis* NCL 912	Continuous cultivation method	Fermentation medium with glucose, yeast extract, soy peptone, MnSO_4_, Tween 80 and MSG, initial pH 5.0. incubated at 32 °C with 150 rpm agitation.	5.11 g/L	[91]
*L. brevis* NCL 912	Fed-batch fermentation	Seed medium containing glucose, soya peptone, MnSO_4_, 4H_2_O, l-glutamate. Incubated at 32 °C for 84 h with initial pH 5.0	103.72 g/L	[92]
*L. brevis* RK03	Cell immobilization with hydrogels 2-hydroxyethyl methacrylate/polyethylene glycol diacrylate (HEMA/PEGDA)	MRS medium containing 450 mM MSG, incubated for 84 h at 30 °C.	39.7 g/L	[93]
*L. brevis* GABA 057	Cell immobilization with alginate beads + isomaltooligosaccharide	GYP medium (pH 4.5) containing MSG incubated for 48 h at 37 °C.	23 g/L	[94]
*L. lactis*	optimizing fermentative condition (temperature 31.9 °C, pH 7.1, 15 g/L of MSG)	Growth on optimized MRS medium containing brown rice, germinated soy bean and skim milk.	7.2 g/L	[34]
*L. brevis* CRL 1942	optimizing culture conditions (30 °C, 48 h, 270 mM MSG)	Growth on optimized MRS medium	26.30 g/L	[96]
*E. faecium* JK29	optimizing MRS medium (0.5% sucrose, 2% yeast extract, 0.5% MSG, pH 7.5, 30 °C)	Growth on optimized MRS medium	1.53 g/L	[38]
*L. brevis HYE1*	optimizing MRS medium (2.14% maltose, 4.01% tryptone, 2.38% MSG, pH 4.74)	Growth on optimized MRS medium	2.21 g/L	[27]
*L. brevis*	modifiying MRS medium containing 6% l-glutamic acid, 4% maltose, 2% yeast extract, 1% NaCl, 1% CaCl_2_, 2 g Tween 80, 0.02 mM PLP, pH 5.25, 37 °C, 72 h	Growth on optimized MRS medium	44.4 g/L	[42]
*L. brevis* Lb85	directed evolution and mutagenesis	Growth on LBG medium supplemented with glucose, kanamycin and l-glutamate, incubated at 30 °C with 200 rpm agitation.	7.13 g/L	[84]
*L. lactis* FJNUGA01	whole-cell bioconversion with pET28a	Resting cells in deionized water with 2 mol/L glutamate, incubated at 45 °C for 6 h	34 g/L	[99]
*L. plantarum* WCFS1	immobilized enzymes to porous silica beads	Enzymatic conversion of 0.5 M MSG, 0.2 mM PLP and 0.02 µg GAD/µL in sodium acetate buffer (pH 5.0), incubated at 37 °C for 20 min.	41.7 g/L	[106]

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
