# Peer review of "Glutamate Decarboxylase from Lactic Acid Bacteria—A Key Enzyme in GABA Synthesis"

_microorganisms, 2020, doi:10.3390/microorganisms8121923_

Round 1
Reviewer 1 Report
Microorganisms – manuscript 995898
A brief review of glutamate decarboxylase from lactic acid bacteria
By
Yogeswara, I. B. A., S. Maneerat, and D. Haltrich
Evaluation
The review presents an overview of the role, function, diversity, and utilization of the glutamate decarboxylase system of lactic acid bacteria. This system involved to cope with acid stress using utilizes extracellular glutamate for an intracellular conversion process yielding gamma-aminobutyric acid (GABA). The decarboxylation activities mediated by LAB are of interest for multiple reasons, including the potential to yield bioactive components such as GABA. Hence the objective of this review was to focus on LAB species that possess the ability to produce GABA through a description of the enzymatic system glutamate decarboxylase (GAD) as well as outline the biochemical properties of GAD. Having an understanding of this information is useful for potential applications, including producing GABA as a bioactive compound and for other commercial applications.
Main comments
Taxonomy. An important part of the review covers description of the genes encoding for the GAD and their diversity between different lactic acid bacteria. Considering the current proposition to redefine the taxonomy of this group of organisms (see https://doi.org/10.1099/ijsem.0.004107), it would have been a nice addition to the text to have a deeper analysis of the relationships between the phylogenetic classification and the functional classification of GAD. It there some other trends between the presence of gadA and/or gadB in relation to the update taxonomical classification or to other functional distinction within the Lactobacillales.
When describing sequences: you need to be more specific as to whether you’re referring to genome sequences or proteinsequences. There were multiple areas in the review where it is unclear as to whether genome sequences or protein sequences were being referenced. See my specific comments under sentence revisions, which are outlined line by line. (line 151, line 153, 284,…). I assume your referring to the protein sequence of the enzymes themselves after expression, in which case this sentence could be reworded as: “Typically, L. brevis contains two GAD-encoding genes, gadA and gadB, which when expressed yield GAD enzymes that share approximately 50% amino acid sequence similarity.”
Biochemical pathways of glutamate, GABA, and beyond. It would be a nice addition to the review to include a figure summarizing the different pathways toward the synthesis of GABA and of the different chemical compounds produced from GABA.
Specific comments
Line 29 - Replace “non-spore forming bacteria” by “non-spore forming bacteria, morphology either cocci or rods…”
Line 41 - This is a run-on sentence, please consider revising.
Line 212 - The abbreviation for pyridoxal 5’-phosphate (PLP) was already defined in a previous section.
Line 214 - Remove “at” before “seven different…” Sentence generally not clear
Line 219 – Sentence difficult to read, need to be rewrite
Line 238 - Place a period after “aldimine”, then start a new sentence. “…as an external aldimine. This Schiff base can then be…”
Line 273 – Missing citation for Wu et al.
Line 343 - Same as line 212, the abbreviation PLP has already been defined.
Line 349 – What do you mean by “…to a medium of S. thermophilus GABA production…”
Line 359 - This sentence should be reworded for clarity. I recommend the following revision by splitting into two sentences: “An engineered strain was constructed by 1) ….., 2) by …. , 3) by ……, and 4) by ….. from the E. coligenome [96]. This resulting strain achieved a productivity of 44.04 g per L of GABA per h with an almost quantitative conversion of 3 M glutamate [96].
Line 371 – Are you referring to C. glutamicum ? Please correct if required.
Line 389 - Reword this sentence to: “A number of immobilization techniques have….”
Figure 1 - Mention in the description that the analysis is from the amino acid sequences
Table 2 – For the column “Effect of various metal ions”, it can be suggest to separate in two columns with one for “increased activity” and the other for “deceased activity” of the ions.
Overall, this review is very thorough and well-prepared. The collection of information is not only robust and current, but they are also very insightful for the topic. Nicely done.
Author Response
Response to reviewers
Reviewer #1
Taxonomy. An important part of the review covers description of the genes encoding for the GAD and their diversity between different lactic acid bacteria. Considering the current proposition to redefine the taxonomy of this group of organisms (see https://doi.org/10.1099/ijsem.0.004107), it would have been a nice addition to the text to have a deeper analysis of the relationships between the phylogenetic classification and the functional classification of GAD. It there some other trends between the presence of gadA and/or gadB in relation to the update taxonomical classification or to other functional distinction within the Lactobacillales.
We appreciate this suggestion – however a first analysis did not show a conclusive relation between the phylogenetic classification and the functional classification of GAD. Maybe a more detailed can reveal such a relationship, but this is currently maybe beyond the scope of this review
When describing sequences: you need to be more specific as to whether you’re referring to genome sequences or protein sequences. There were multiple areas in the review where it is unclear as to whether genome sequences or protein sequences were being referenced. See my specific comments under sentence revisions, which are outlined line by line. (line 151, line 153, 284,…). I assume your referring to the protein sequence of the enzymes themselves after expression, in which case this sentence could be reworded as: “Typically, L. brevis contains two GAD-encoding genes, gadA and gadB, which when expressed yield GAD enzymes that share approximately 50% amino acid sequence similarity.”
was checked and corrected (e.g, the mentioned sentence was corrected as suggested, and the entire paragraph was corrected to distinguish between gene and protein)
Biochemical pathways of glutamate, GABA, and beyond. It would be a nice addition to the review to include a figure summarizing the different pathways toward the synthesis of GABA and of the different chemical compounds produced from GABA.
several other reviews on GABA production show these pathways in detail, so it might not be necessary to add it to this review focusing on GAD as well
Specific comments
Line 29 - Replace “non-spore forming bacteria” by “non-spore forming bacteria, morphology either cocci or rods…”
done
Line 41 - This is a run-on sentence, please consider revising.
done
Line 212 - The abbreviation for pyridoxal 5’-phosphate (PLP) was already defined in a previous section.
done
Line 214 - Remove “at” before “seven different…” Sentence generally not clear
done, sentence has been rephrased
Line 219 – Sentence difficult to read, need to be rewrite
done, sentence has been rephrased
Line 238 - Place a period after “aldimine”, then start a new sentence. à “…as an external aldimine. This Schiff base can then be…”
done, sentence has been rephrased
Line 273 – Missing citation for Wu et al.
done, citation has been added
Line 343 - Same as line 212, the abbreviation PLP has already been defined.
done
Line 349 – What do you mean by “…to a medium of S. thermophilus GABA production…”
done, sentence has been rephrased, it now says “when L-glutamate was added at concentrations of 10 to 20 g/L to the growth medium of S. thermophilus, GABA production could not be enhanced”
Line 359 - This sentence should be reworded for clarity. I recommend the following revision by splitting into two sentences: “An engineered strain was constructed by 1) ….., 2) by …. , 3) by ……, and 4) by ….. from the E. coli genome [96]. This resulting strain achieved a productivity of 44.04 g per L of GABA per h with an almost quantitative conversion of 3 M glutamate [96].
done, this sentences was modified to “An engineered strain was constructed by (i) introducing mutations into this GadB to shift its decarboxylation activity toward a neutral pH, (ii) by modifying the glutamate/GABA antiporter GadC to facilitate transport at a neutral pH, (iii) by enhancing the expression of soluble GadB through overexpression of the GroESL molecular chaperones, and (iv) by inhibiting the degradation of GABA through inactivation of gadA and gadB from the E. coli genome. This resulting engineered strain achieved a productivity of 44.04 g per L of GABA per h with an almost quantitative conversion of 3 M glutamate [97].“
Line 371 – Are you referring to C. glutamicum ? Please correct if required.
done, we are referring to C. glutamicum of course
Line 389 - Reword this sentence to: “A number of immobilization techniques have….”
the sentence has now been rephrased to “A number of immobilization techniques have been applied for re-use of the biocatalyst, such as immobilization of GadB…”
Figure 1 - Mention in the description that the analysis is from the amino acid sequences
done, the legend now says “Phylogenetic analysis of glutamate decarboxylase from different species of LAB. The phylogenetic tree was calculated based on the amino acid sequences of GAD (maximum-likelihood method). The phylogenetic analysis was performed after the alignment of GAD sequences using MUSCLE in MEGA X software.“
Table 2 – For the column “Effect of various metal ions”, it can be suggest to separate in two columns with one for “increased activity” and the other for “deceased activity” of the ions.
done
Overall, this review is very thorough and well-prepared. The collection of information is not only robust and current, but they are also very insightful for the topic. Nicely done.
Thank you very much!

Reviewer 2 Report
In this review manuscript, the authors summarize recent research about glutamate decarboxylase (GAD) found in lactic acid bacteria (LAB) with following contests:
- Diversity of GABA-producing LAB
- Occurrence of GAD genes
- Catalytic mechanism and biochemical properties of GAD
- Improvement of GAD activities through mutations and GABA productivity by bioengineering
- The role in industry and future insight of GABA production by GAD
As a first impression, the title of the manuscript is not informative and representative of manuscript contents. I felt that this review focused on GABA production as well as GAD itself. Therefore, the title should be reconsidered, maybe addition of a subtitle is effective. At least a keyword “GABA” or “GABA production” should be included.
In addition, as “a review paper”, this manuscript is not helpful for readers because this manuscript simply lists the results in previous papers, and lacks the analysis, discussion, or explanation of previous results by the author. Efforts are needed to support the readers’ understanding with figures and tables in each chapter.
Overall, I don’t understand the concept of the manuscript (as the impression that the title and contents are inconsistent), therefore in its current form this manuscript is not suitable for publication.
In revision of the manuscript, please consider adding the following information in order to help readers to understand the research filed as a collection of previous GAD researches.
- Reconsider the manuscript title. I felt that “GABA” should be included.
- Citations should be displayed in order. Even citations that were not mentioned in the manuscript body at all were listed.
- The reaction formula of GAD should be included in “Introduction”.
- In “3. Biodiversity of GABA-producing Lactic Acid Bacteria”, consider adding a table about GABA productivity, titer, fermentation condition, etc. of reported LAB, similar to Table 2. Such list should be a good information for GABA producing LAB.
- In “4. Glutamate dehydrogenase” section, the figures which represent the active center, crystal structure, reaction mechanism, and mutation residues of GAD. Such figures will support the understanding of this section.
- “5. Biochemical -----” section. This section also required the figure of the structure and the active center of GAD.
- “6. Improvement of -----” section. Please consider adding a table for efforts for improving GABA productivity. It should be much easier to understand in a table than in a sentence.
Author Response
Reviewer #2
In revision of the manuscript, please consider adding the following information in order to help readers to understand the research filed as a collection of previous GAD researches.
- Reconsider the manuscript title. I felt that “GABA” should be included.
We now changed the title to “Glutamate Decarboxylase from Lactic Acid Bacteria - a Key Enzyme in GABA Synthesis” so that GABA is included as a key word - Citations should be displayed in order. Even citations that were not mentioned in the manuscript body at all were listed.
we checked and corrected the citations - The reaction formula of GAD should be included in “Introduction”.
we added a new Figure 1 which shows the decarboxylation reaction catalysed by GAD - In “3. Biodiversity of GABA-producing Lactic Acid Bacteria”, consider adding a table about GABA productivity, titer, fermentation condition, etc. of reported LAB, similar to Table 2. Such list should be a good information for GABA producing LAB.
GABA production in LAB (together with its physiological effects) has been reviewed several times recently, and our aim of this review was to focus mainly on GAD itself and not so much on GABA; to acknowledge this valid point we added a sentences referring to these recent reviews at the end of this section; this sentences runs as “Production of GABA by different LAB together with fermentation conditions, yields and productivities has been recently reviewed [15,43,48].“ - In “4. Glutamate dehydrogenase” section, the figures which represent the active center, crystal structure, reaction mechanism, and mutation residues of GAD. Such figures will support the understanding of this section.
we now added two new figures to this section, showing the GAD monomer with its covalently attached PLP as well as the residues discussed in this section, i.e. residues involved in substrate interaction as well as catalytic residues (close-up of the active site) - “5. Biochemical -----” section. This section also required the figure of the structure and the active center of GAD.
we added one new figure to this section showing the dimeric architecture of GAD; a new figure showing the active site is part of the section ‘glutamate decarboxylase’ (new Figure 5) - “6. Improvement of -----” section. Please consider adding a table for efforts for improving GABA productivity. It should be much easier to understand in a table than in a sentence.
we added a new table 3, showing various approaches to improvement of GABA productivity

Round 2
Reviewer 2 Report
Thank you for addressing my comments. I recommend acceptance of this manuscript after addressing following minor points.
- In Figures 1 and 2, chemical structures of GABA and PLP are not suitable.
GABA structure should be replaced by structure with ChemDraw format (or other similar software). One of -OH group of phosphate moiety of PLP should be corrected (Correct "-HO" to "-OH").
- References must be numbered in order of appearance in the text as "Introduction for Authors".
For example,
Line 111, “[2, 13, 31, 32, 33, 42, 47]”. Refs. 33-41, 43-46 were not numbered in order.
In Table 1, Refs. 47 and 95 were also not numbered in order.
Please recheck and correct the citations.
- Table 3. I think the Table 3 is useful.
Could you include following information in Table 3?
-
- Distinguish between fermentation and resting-cell reaction
- Reaction conditions (substrate or carbon source or substrate, reaction time)
The unit of GABA production can be united?? Even if the GABA production was reported by mM in the original paper, it can be changed to g/L.
Author Response
Reviewer #2
- In Figures 1 and 2, chemical structures of GABA and PLP are not suitable.
GABA structure should be replaced by structure with ChemDraw format (or other similar software). One of -OH group of phosphate moiety of PLP should be corrected (Correct "-HO" to "-OH").
Response: Done, we already changed the figures. We changed the HO to OH of phosphate group moiety of PLP
- References must be numbered in order of appearance in the text as "Introduction for Authors".
For example,
Line 111, “[2, 13, 31, 32, 33, 42, 47]”. Refs. 33-41, 43-46 were not numbered in order.
In Table 1, Refs. 47 and 95 were also not numbered in order.
Please recheck and correct the citations.
Response: Done, we already recheck the citations and numbered in order.
- Table 3. I think the Table 3 is useful.
Could you include following information in Table 3?
- Distinguish between fermentation and resting-cell reaction
- Reaction conditions (substrate or carbon source or substrate, reaction time)
The unit of GABA production can be united?? Even if the GABA production was reported by mM in the original paper, it can be changed to g/L.
Response: Done, we add an additional information for Table 3